# Genomic and Molecular Identification of Genes Contributing to the Caspofungin Paradoxical Effect in *Aspergillus fumigatus*

Shu Zhao,[a,b] Adela Martin-Vicente,[c] Ana Cristina Colabardini,[d] Lilian Pereira Silva,[d] David C. Rinker,[e] Jarrod R. Fortwendel,[c] Gustavo Henrique Goldman,[d] John G. Gibbons[a,b,f]

[a]Molecular and Cellular Biology Graduate Program, University of Massachusetts, Amherst, Massachusetts, USA

[b]Department of Food Science, University of Massachusetts, Amherst, Massachusetts, USA

[c]Department of Clinical Pharmacy and Translational Science, University of Tennessee Health Science Center, Memphis, Tennessee, USA

[d]Faculdade de Ciências Farmacêuticas de Ribeirão Preto, Universidade de São Paulo, São Paulo, Brazil

[e]Department of Biological Sciences, Vanderbilt University, Nashville, Tennessee, USA

[f]Organismic and Evolutionary Biology Graduate Program, University of Massachusetts, Amherst, Massachusetts, USA

Shu Zhao and Adela Martin-Vicente contributed equally to this article. Author order was determined by the length of involvement in the study.

**ABSTRACT** *Aspergillus fumigatus* is a deadly opportunistic fungal pathogen responsible for ~100,000 annual deaths. Azoles are the first line antifungal agent used against *A. fumigatus*, but azole resistance has rapidly evolved making treatment challenging. Caspofungin is an important second-line therapy against invasive pulmonary aspergillosis, a severe *A. fumigatus* infection. Caspofungin functions by inhibiting $\beta$-1,3-glucan synthesis, a primary and essential component of the fungal cell wall. A phenomenon termed the caspofungin paradoxical effect (CPE) has been observed in several fungal species where at higher concentrations of caspofungin, chitin replaces $\beta$-1,3-glucan, morphology returns to normal, and growth rate increases. CPE appears to occur *in vivo*, and it is therefore clinically important to better understand the genetic contributors to CPE. We applied genomewide association (GWA) analysis and molecular genetics to identify and validate candidate genes involved in CPE. We quantified CPE across 67 clinical isolates and conducted three independent GWA analyses to identify genetic variants associated with CPE. We identified 48 single nucleotide polymorphisms (SNPs) associated with CPE. We used a CRISPR/Cas9 approach to generate gene deletion mutants for seven genes harboring candidate SNPs. Two null mutants, ΔAfu3g13230 and ΔAfu4g07080 (*dscP*), resulted in reduced basal growth rate and a loss of CPE. We further characterized the *dscP* phosphatase-null mutant and observed a significant reduction in conidia production and extremely high sensitivity to caspofungin at both low and high concentrations. Collectively, our work reveals the contribution of Afu3g13230 and *dscP* in CPE and sheds new light on the complex genetic interactions governing this phenotype.

**IMPORTANCE** This is one of the first studies to apply genomewide association (GWA) analysis to identify genes involved in an *Aspergillus fumigatus* phenotype. *A. fumigatus* is an opportunistic fungal pathogen that causes hundreds of thousands of infections and ~100,000 deaths each year, and antifungal resistance has rapidly evolved in this species. A phenomenon called the caspofungin paradoxical effect (CPE) occurs in some isolates, where high concentrations of the drug lead to increased growth rate. There is clinical relevance in understanding the genetic basis of this phenotype, since caspofungin concentrations could lead to unintended adverse clinical outcomes in certain cases. Using GWA analysis, we identified several interesting candidate polymorphisms and genes and then generated gene deletion mutants to determine whether these genes were important for CPE. Two of these mutant strains (ΔAfu3g13230 and ΔAfu4g07080/Δ*dscP*) displayed a loss of the CPE. This study sheds light on the genes involved in clinically important phenotype CPE.

Address correspondence to Jarrod R. Fortwendel, jfortwen@uthsc.edu, Gustavo Henrique Goldman, ggoldman@usp.br, or John G. Gibbons, jggibbons@umass.edu.

The authors declare no conflict of interest.

**KEYWORDS** *Aspergillus fumigatus*, CRISPR, echinocandin, GWA, genomics, mycology, caspofungin

The genus *Aspergillus* encompasses more than 340 saprophytic filamentous fungal species that can grow over broad temperature and pH ranges (1). *Aspergillus* species can be isolated from different environments and can have beneficial and/or detrimental effects to humans (2, 3). For instance, some species are used in food fermentation, as well as in the industrial production of enzymes, organic acids, bioactive compounds, and pharmaceuticals (4–8). Conversely, other *Aspergillus* species are common agents of food spoilage, and others negatively affect food security through the production of mycotoxins, causing economic losses in agricultural commodities and serious health problems (9–11). In addition, a small number of well-characterized *Aspergillus* species are regarded as human pathogens, with *Aspergillus fumigatus* being responsible for more than 90% of infections (12). In healthy individuals, *Aspergillus* infections can lead to noninvasive forms of infection, ranging from colonization of a parenchymal lung cavity (aspergilloma) to chronic inflammatory and fibrotic process (chronic pulmonary aspergillosis) (13). Atopic patients may experience hypersensitivity after exposure to *A. fumigatus* allergens, a pulmonary disorder called allergic bronchopulmonary aspergillosis, which complicates the course of asthma and cystic fibrosis (14). In severely immunocompromised individuals, the lung colonization may be followed by dissemination to other organs, a condition known as invasive pulmonary aspergillosis (IPA) (15).

The frequency of IPA has increased substantially in the last decades due to increased survival of high-risk populations. Most IPA cases occur in patients with hematological malignancies (16, 17), followed by patients treated in intensive care units (18), individuals receiving immunosuppressive therapy (19), and individuals receiving solid organ transplants (20, 21). The morbidity and mortality rates associated with IPA are extremely high, reaching >90% in cases where the brain is affected (22). Recently, aspergillosis has worsened the burdens of the COVID-19 pandemic, since many COVID-19 patients are infected by *A. fumigatus* (23–29).

The main classes of antifungal drugs used to treat IPA include polyenes, azoles, and echinocandins. Polyenes, such as amphotericin B, sequesters ergosterol from the cell membrane (30), while azoles inhibit ergosterol biosynthesis (31). Ergosterol plays an essential functional role in regulating cell membrane permeability and fluidity. Echinocandins, such as caspofungin, disrupt the biosynthesis of $\beta$-1,3-glucan, an essential component of the fungal cell wall (32). Azoles are the first-line therapeutic agents against IPA, while caspofungin represents the second-line therapy. Caspofungin acts by noncompetitive inhibition of $\beta$-1,3-glucan synthesis, the main component of the fungal cell wall, which results in growth inhibition and increased osmotic sensitivity (33). Echinocandins have only a fungistatic effect on filamentous fungi; however, echinocandin use is gaining interest because *A. fumigatus* has rapidly evolved resistance mechanisms to azoles and because of the limitations related to drug interactions and/or toxicity with azoles and polyenes (34).

Caspofungin exposure modifies the composition and organization of the *A. fumigatus* cell wall, resulting in hyperbranching, lysis of hyphal apical compartments, loss of cell wall $\beta$-1,3-glucan, and chitin overproduction (35). However, at higher drug concentrations, long hyphae with normal morphology, reconstituted $\beta$-1,3-glucan synthesis, and normalized levels of cell wall chitin emerge from the slow-growing microcolonies (36). This phenomenon, called the caspofungin paradoxical effect (CPE), also appears to exist *in vivo*, although its clinical relevance is not well understood (37, 38). The precise mechanism behind CPE appears to depend on a complex network of interactions between components of different pathways that work together to reestablish *A. fumigatus* growth (39). The initial trigger of CPE consists of the increase in intracellular $Ca^{2+}$, which binds to calmodulin and activates the phosphatase calcineurin (40). Active calcineurin dephosphorylates specific transcription factors that regulate the activation of several stress responses and cell wall modifications (41–43). $Ca^{2+}$ deprivation (44) and the inhibition of either heat shock protein 90 or the mitochondrial respiratory chain (43, 45) result in the

abolition of CPE in *A. fumigatus*. Conversely, mitochondrial ROS accumulation (in response to caspofungin exposure) alters the plasma membrane lipid composition, causing a conformational change in the Fks1 enzyme, which likely prevents caspofungin binding and therefore restores $\beta$-1,3-glucan synthase activity (46).

In addition, a role of the cell wall integrity (CWI) pathway was indicated for its involvement in CPE. The CWI mitogen-activated protein kinase MpkA and its associated transcription factor RlmA regulate chitin synthase gene expression and positively impact the expression of genes involved in $\beta$-1,3-glucan and $\alpha$-1,3-glucan biosynthesis in response to different concentrations of caspofungin (47). Moreover, the SakA mitogen-activated protein kinase of the high-osmolarity glycerol pathway is also activated during cell wall stress and contributes to MpkA activation during adaptation to caspofungin stress (39). Overall, the signaling pathways involved in CPE are not fully understood in *A. fumigatus*, so the exact molecular mechanisms and interactions between the components of these pathways have not yet been fully elucidated.

Genomewide association (GWA) analysis has emerged as an effective tool for discovering genetic variants and genes associated with complex phenotypes. For instance, GWA has been applied to study the genetic basis of fungal phenotypes (48–54), including drug sensitivity and tolerance in *A. fumigatus* (55–58). Here, we performed GWA analysis for CPE in 67 clinical isolates of *A. fumigatus* to provide insight into the variants, genes and pathways contributing to CPE.

## RESULTS

**Quantification of CPE across *A. fumigatus* strains.** We quantified CPE by calculating the recovery rate for 67 *A. fumigatus* clinical isolates. Twenty-six strains did not show CPE (CPE⁻), while 41 possessed the CPE phenotype (CPE⁺) (Fig. 1; see also Table S1 in the supplemental material). We first tested whether CPE⁻ and CPE+ strains differed in growth rate at various concentrations of caspofungin (0, 0.125, 0.25, 0.5, 1, and 8 $\mu$g/mL). After applying a Bonferroni multiple-test correction $P$ value cutoff of 0.0083, the only significant growth difference observed between CPE⁻ and CPE⁺ isolates occurred at 8 $\mu$g/mL caspofungin, where, as expected, CPE⁺ isolates grew faster (see Fig. S1).

**Population structure of CPE isolates.** We used PCA to examine the population structure of the 67 *A. fumigatus* isolates. We identified four distinct *A. fumigatus* genetic clusters (Fig. 2). Cluster A was the most distant and separated from clusters B to D on PC1, while clusters B, C, and D separated on PC2 (Fig. 2). PC1 and PC2 explained 67.9% of variance. Phylogenetic network analysis further confirmed the relationship between isolates (see Fig. S2). Next, we performed a $\chi^2$ statistical analysis to test the null hypothesis that CPE⁺ and CPE⁻ isolates were evenly distributed across the four populations. Statistical analysis supported rejecting the null hypothesis ($\chi^2$ = 11.34, df = 3, $P$ = 0.01). Deviation of observed versus expected frequencies of CPE⁺ and CPE⁻ derives from populations A and B which show greater frequencies of CPE⁻ and CPE⁺ isolates, respectively (Fig. 2; see also Fig. S3).

**Genomewide association for CPE.** To identify associations between genetic variants and CPE, we conducted GWA analysis using 181,309 single nucleotide polymorphisms (SNPs). We conducted three independent GWA analyses. In all analyses, we examined the SNPs with the 25 lowest $P$ values. We also generated quantile-quantile (Q-Q) plots of expected versus observed $P$ values to inspect $P$ value inflation, which could be the product of inadequate correction of population structure. The Q-Q plots indicated that the distribution of $P$ values for each of the GWA analyses were not inflated (see Fig. S4).

First, we performed GWA analyses in all isolates, using a linear model and four PCs to correct for population structure in PLINK (59). In this analysis, we considered $P$ values of <0.00044 as significant (Fig. 3A and Table 1). We identified 2, 12, 7, 2, 1, and 2 SNPs associated with CPE on chromosomes 2, 3, 4, 5, 6, and 7, respectively. These SNPs overlapped 21 genes and included 6 SNPs located in upstream regions, 5 SNPs located in 5′ untranslated regions (UTRs), 7 missense variants, 6 synonymous variants, and 2

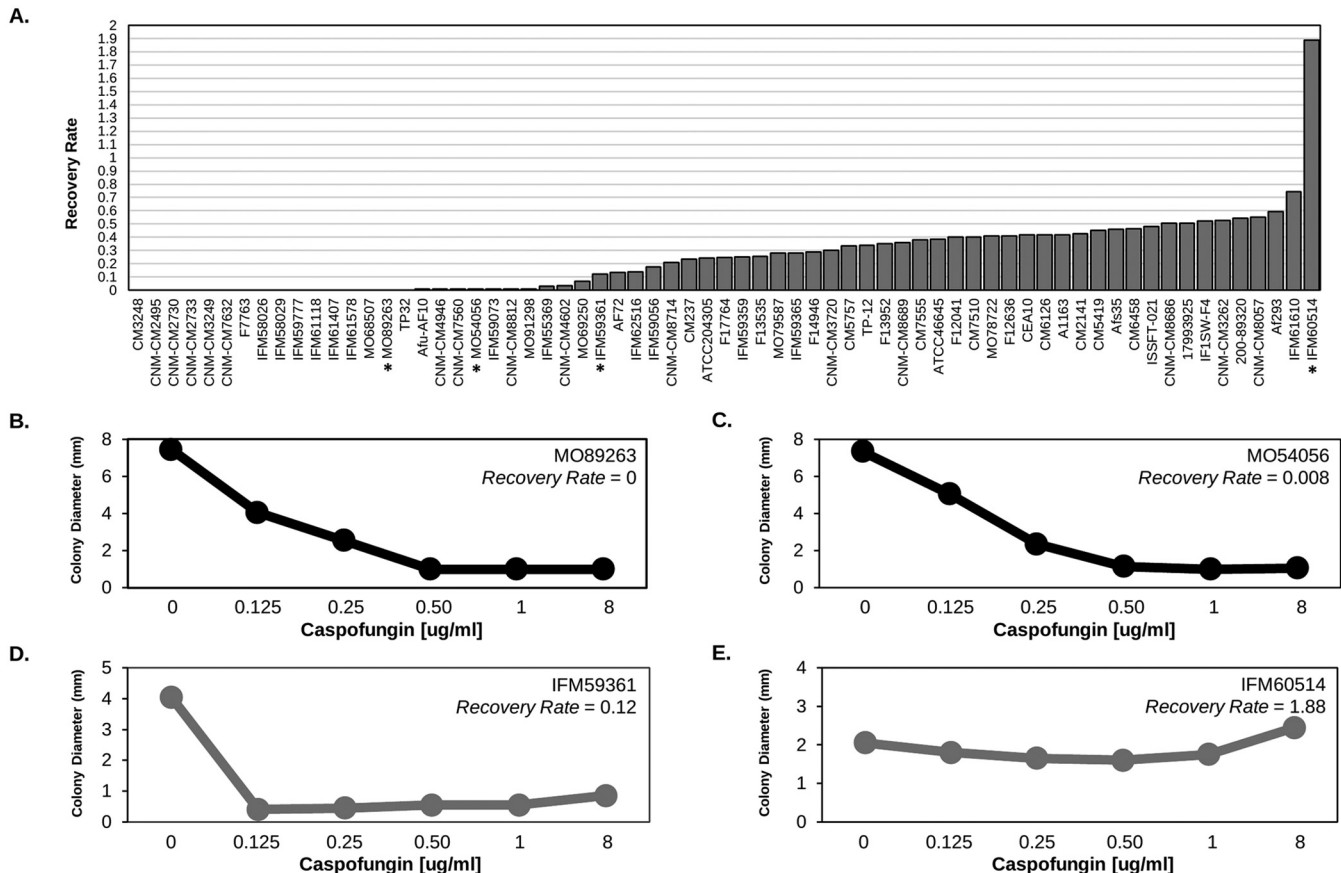

**FIG 1** Quantification of CPE across *A. fumigatus* isolates. (A) CPE, as measured by the recovery rate (*y* axis), was calculated across the 67 *A. fumigatus* isolates (*x* axis). We considered isolates displaying a recovery rate of ≥0.1 as possessing the CPE. An asterisk (*) under the isolate identifier represents isolates in panels B to D. (B to E) Examples of isolates lacking (B and C) and possessing (D and E) the CPE phenotype. The average colony diameter (*y* axis) is plotted for caspofungin concentrations of 0, 0.125, 0.25, 0.5, 1, and 8 $\mu$g/mL (*x* axis).

splice region/intron variants (Table 1). Of note, we identified an 18.2-kb region on chromosome 3 containing 10 SNPs associated with CPE that overlapped Afu3g13230, Afu3g13250, Afu3g13260, Afu3g13270, and Afu3g13300 (Fig. 3A and Table 1; see also Fig. S5).

Next, we performed GWA analyses with the exclusion of the 11 isolates from the more distantly related population A (Fig. 2). Again, we used a linear GWA model with 4 PC to correct for population structure. We considered *P* values of <0.00021 significant. Of the 25 SNPs with the strongest association with CPE, 19 overlapped with the 67-sample GWA. Four of the six SNPs exclusively associated with CPE in the 56-sample analysis were located near the large region on chromosome 3 identified in the 67-sample analysis. These variants overlapped two additional genes not identified in the 67-sample GWA (a 3′ UTR variant in Afu3g13390 and a synonymous variant in Afu3g13400) (Fig. 3B and Table 1).

Lastly, we conducted GWA analyses with isolates that had the 25 highest recovery rate values (recovery rate > 0.349), and 25 lowest recovery rate values (recovery rate < 0.035). Of the 25 significant SNPs, 9 overlapped with the 67-sample and 56-sample analyses, while 16 SNPs were exclusively identified in the 50-sample analysis. Of these 16 SNPs, 7 were present within the chromosome 3 locus identified in the other GWA analyses, while 6, 1, and 2 SNPs were present on chromosomes 2, 1, and 7, respectively (Fig. 3C and Table 1).

**CRISPR/Cas9 gene deletion and overexpression of candidate genes.** We used a CRISPR/Cas9 based approach to delete seven genes that harbored SNPs that were significantly associated with CPE. Specifically, we knocked out Afu2g08660, which encodes SltB, a component of the SltA-dependent pathway involved in cation homeostasis

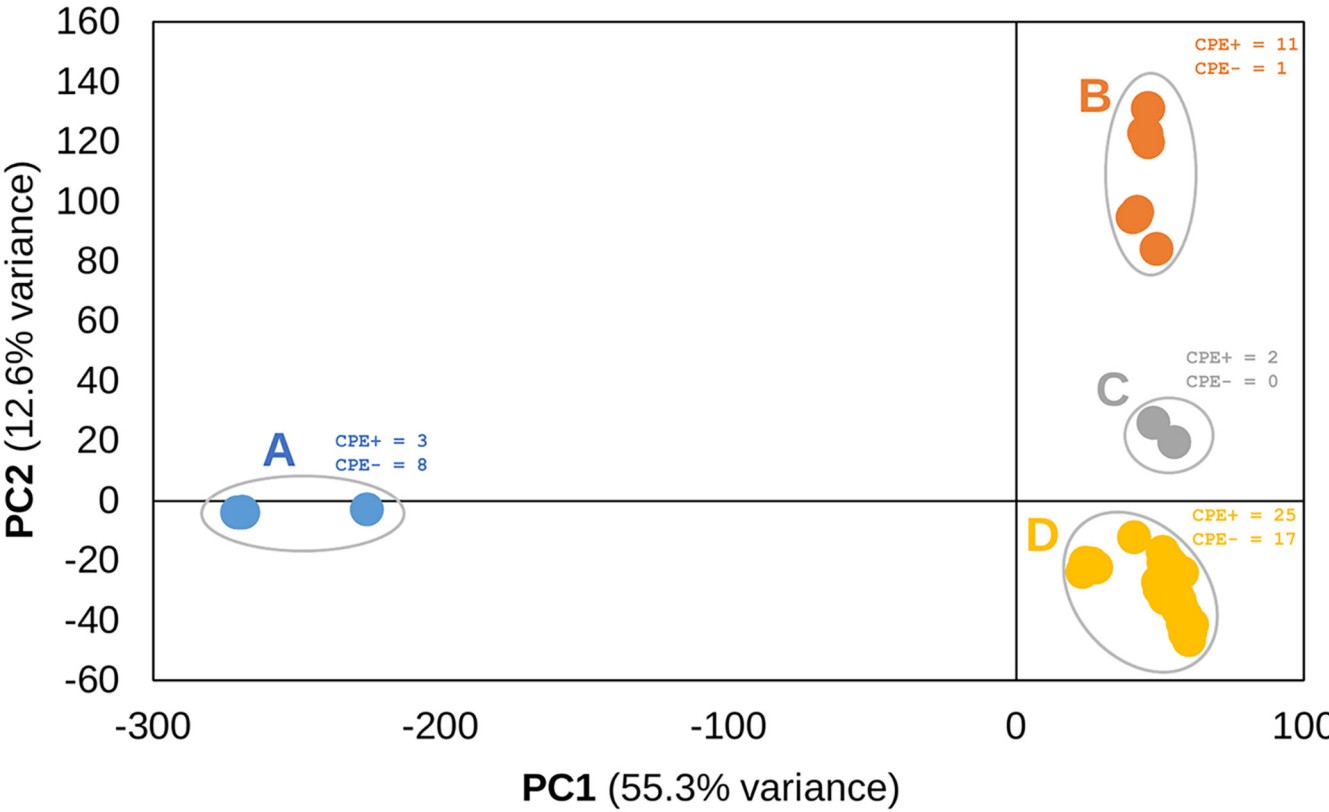

**FIG 2** Population structure of *A. fumigatus* isolates. Principal-component analysis (PCA) of the 67 *A. fumigatus* isolates using 181,309 SNPs. Four major populations are present. Population A separates from populations B to D on PC1, which explains 55.3% variance, while populations B, C, and D separate on PC2, which explains 12.6% of variance. The numbers of CPE+ and CPE− isolates are provided for each population.

(60) (detected in the 67-sample GWA); Afu3g13230, which encodes a hypothetical protein with an AT Hook DNA binding motif (detected in all GWAs); Afu3g13260, which encodes a hypothetical protein with domains with predicted nuclease activity and role in DNA repair (detected in all GWAs); Afu3g13270, which encodes DgkA, a putative diacylglycerol kinase (detected in all GWAs); Afu4g07080, which encodes DspC, a predicted tyrosine phosphatase (61) (detected in the 67-sample GWA); Afu7g01440, which encodes a hypothetical protein (detected in the 56-sample and 67-sample GWAs); and Afu7g01560, which encodes a hypothetical protein (detected in all GWAs). Candidate genes were knocked out by replacement with the *A. parasiticus pyrG* gene in the uracil-auxotrophic *A. fumigatus* Ku80Δ*pyrG* (CEA17) genetic background. Importantly, *A. fumigatus* CEA17, which is a CEA10 derivative, is CPE+ (Fig. 1; recovery rate = 0.42), allowing us to directly test the effect of each gene deletion in the presence of increasing levels of caspofungin. Two to four null mutants were first grown on glucose minimal media (GMM) in the presence of 0, 0.125, and 4 $\mu$g/mL caspofungin, and growth patterns were inspected qualitatively (Fig. 4). The parental strain (Ku80Δ *pyrG*) shows a clear increase in growth rate at a caspofungin concentration of 4 $\mu$g/mL versus 0.125 $\mu$g/mL (Fig. 4 and Fig. 5C). The Δ*dspC* and ΔAfu3g13230 mutants displayed a significant reduction in basal growth rate in the absence of caspofungin (Fig. 5A and B). However, these null mutants also displayed a loss of the CPE phenotype (Fig. 5C and D). The loss of CPE was consistent across independent ΔAfu3g13230 and Δ*dscP* mutants (see Fig. S6).

Because null expression of Afu3g13230 and *dscP* resulted in a loss of the CPE phenotype, in addition to a slow growth phenotype, we also tested whether overexpression would result in maintenance of the CPE phenotype or an exaggerated CPE phenotype. We generated two independent overexpression mutants for Afu3g13230 and three independent overexpression mutants for *dscP* by replacing the exogenous promoters with the *hspA* promoter (62). The overexpression mutant phenotypes for both genes were

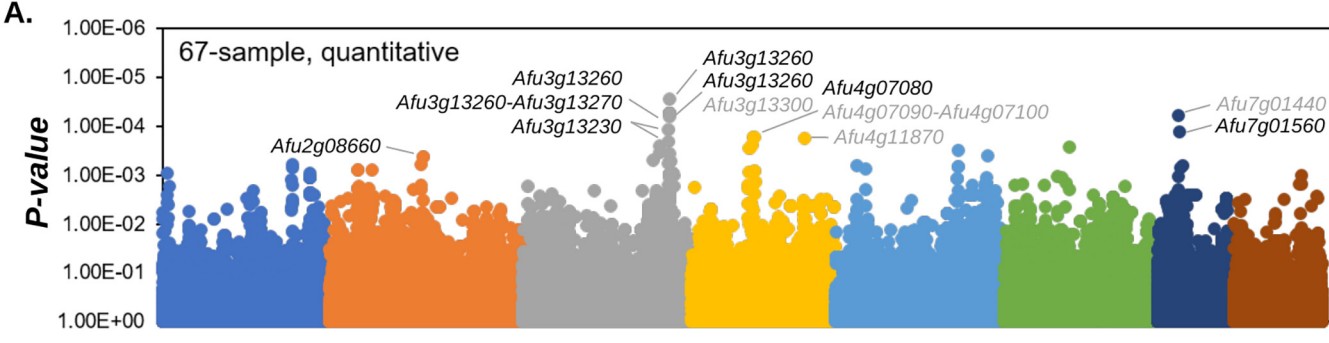

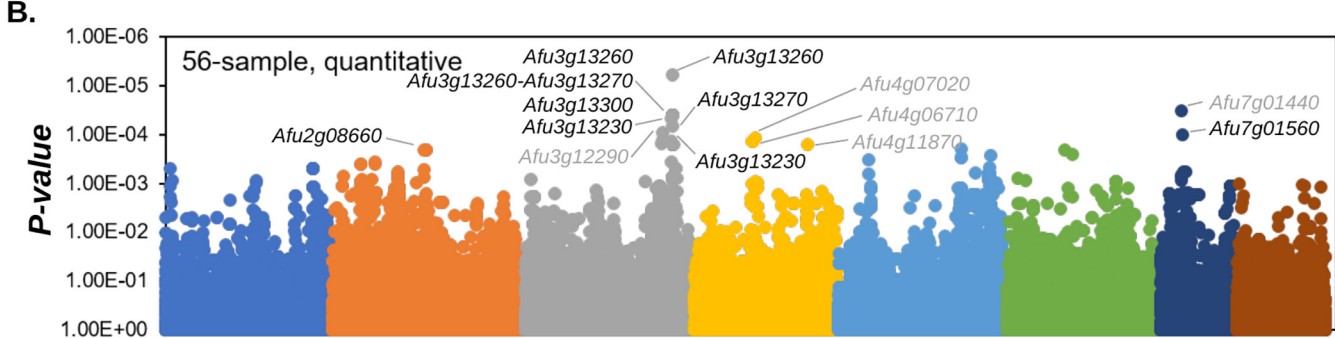

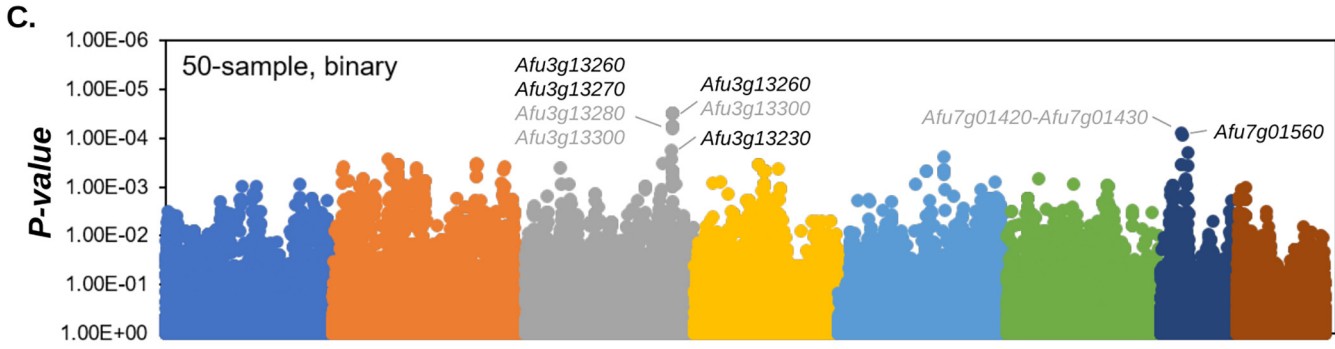

**FIG 3** Genomewide association for CPE. Manhattan plots for the 67-sample (A), 56-sample (B), and 50-sample (C) GWAs are shown. Genes overlapping the lowest 25 *P* values for each GWA are labeled. Gene labels in black represent genes that were experimentally knocked out. Colors represent chromosomes 1 to 8.

highly similar to the wild type (i.e., maintenance of the CPE phenotype) (see Fig. S7), which suggests the expression of these genes are necessary for CPE.

**Phenotypic characterization of the Δ*dscP* mutant.** We phenotypically characterized the Δ*dscP* strain because its growth patterns differed drastically in comparison to the parental strain (Fig. 4 and 5). First, we quantified relative conidia production (conidia per mm² of colony diameter) after growth on GMM at 37°C for 96 h. We observed a significant reduction in conidia production in the Δ*dscP* strain ($P < 0.0001$) (Fig. 6A). Next, ~1,000 conidia from the parental and Δ*dscP* strains were inoculated onto coverslips, grown in liquid GMM, and stained with Calcofluor White (CFW) to visualize impairment of hyphal morphogenesis with and without echinocandin stress. Our results suggest that the Δ*dscP* strain is highly sensitive to caspofungin at both relatively low (0.125 μg/mL) and high concentrations (4 μg/mL), since both caused complete loss of hyphal formation underpinned by germling tip lysis in this mutant (Fig. 6B). In contrast, the control strain was able to generate microcolonies consisting of hyphae with a mixture of lysed and intact hyphal tips at both concentrations of drug (Fig. 6B). Collectively, these results suggest that DspC contributes to asexual development, hyphal growth, and the response to caspofungin stress.

**TABLE 1** Characterization of SNPs associated with CPE[a]

| Chr | Pos | Ref | Alt | Gene | SnpEff annotation | $P$[b] | | |
|---|---|---|---|---|---|---|---|---|
| | | | | | | P-val_67 | P-val_56 | P-val_50 |
| Chr2 | 1182496 | G | A | Afu2g04280 | upstream_gene_variant | 0.008605 | 0.005486 | **0.0002763** |
| Chr2 | 2226667 | T | C | Afu2g08660 | synonymous_variant | **0.0004305** | 0.0002084 | 0.9817 |
| Chr2 | 2228799 | A | G | Afu2g08670 | missense_variant | **0.0004305** | 0.0002084 | 0.9817 |
| Chr2 | 3620195 | A | C | Afu2g13870 | upstream_gene_variant | 0.008667 | 0.004488 | **0.0003462** |
| Chr2 | 3620629 | A | G | Afu2g13870 | 5_prime_UTR_variant | 0.005414 | 0.003009 | **0.0003462** |
| Chr2 | 3621501 | C | T | Afu2g13870 | missense_variant | 0.009206 | 0.004488 | **0.0003242** |
| Chr2 | 3622643 | T | G | Afu2g13880 | upstream_gene_variant | 0.008667 | 0.004488 | **0.0003462** |
| Chr2 | 3623412 | G | T | Afu2g13880 | synonymous_variant | 0.005414 | 0.003009 | **0.0003462** |
| Chr3 | 3220395 | G | A | Afu3g12220 | missense_variant | 0.002289 | **0.0001545** | 0.975 |
| Chr3 | 3242018 | T | C | Afu3g12290 | 5_prime_UTR_variant | **0.0002593** | 9.25E−05 | 0.9782 |
| Chr3 | 3242454 | A | C | Afu3g12300 | 5_prime_UTR_variant | **0.0003515** | 0.0001318 | 0.0003255 |
| Chr3 | 3511190 | C | T | Afu3g13230 | missense_variant | **0.0001198** | 4.59E−05 | **0.000176** |
| Chr3 | 3512039 | G | A | Afu3g13230 | missense_variant | 0.002489 | 0.001453 | **0.0002727** |
| Chr3 | 3514244 | T | C | Afu3g13230 | upstream_gene_variant | **0.000203** | **0.0001363** | 0.00843 |
| Chr3 | 3519307 | G | A | Afu3g13260 | synonymous_variant | 0.0005377 | **0.0001639** | 0.9823 |
| Chr3 | 3519801 | T | A | Afu3g13260 | missense_variant | 0.0009737 | 0.000746 | **5.88E−05** |
| Chr3 | 3520297 | G | A | Afu3g13260 | synonymous_variant | **5.41E−05** | **4.00E−05** | **3.06E−05** |
| Chr3 | 3520699 | G | T | Afu3g13260 | synonymous_variant | **2.86E−05** | **5.95E−06** | 0.981 |
| Chr3 | 3521662 | A | C | Afu3g13250 | upstream_gene_variant | **5.41E−05** | **4.00E−05** | **3.06E−05** |
| Chr3 | 3522032 | T | G | Afu3g13270 | missense_variant | 0.001099 | 0.000746 | **6.34E−05** |
| Chr3 | 3522988 | C | G | Afu3g13270 | 5_prime_UTR_variant | **0.0003703** | 6.82E−05 | 0.0009726 |
| Chr3 | 3525646 | C | A | Afu3g13280 | 3_prime_UTR_variant | 0.0009737 | 0.000746 | **5.88E−05** |
| Chr3 | 3527933 | A | T | Afu3g13300 | splice_region_variant&intron_variant | **5.41E−05** | **4.00E−05** | **3.06E−05** |
| Chr3 | 3527935 | A | T | Afu3g13300 | splice_region_variant&intron_variant | **5.41E−05** | **4.00E−05** | **3.06E−05** |
| Chr3 | 3528532 | C | T | Afu3g13300 | synonymous_variant | 0.0009737 | 0.000746 | **5.88E−05** |
| Chr3 | 3528767 | C | T | Afu3g13300 | missense_variant | 0.0005377 | **0.0001639** | 0.9823 |
| Chr3 | 3529138 | A | G | Afu3g13300 | missense_variant | **6.31E−05** | **4.00E−05** | **3.19E−05** |
| Chr3 | 3529460 | G | C | Afu3g13300 | missense_variant | **5.41E−05** | **4.00E−05** | **3.06E−05** |
| Chr3 | 3529903 | G | A | Afu3g13300 | missense_variant | 0.0009737 | 0.000746 | **5.88E−05** |
| Chr3 | 3530260 | G | A | Afu3g13300 | missense_variant | 0.0009737 | 0.000746 | **5.88E−05** |
| Chr3 | 3543581 | T | A | Afu3g13390 | 3_prime_UTR_variant | 0.0005119 | **0.0001583** | 0.006607 |
| Chr3 | 3546410 | C | T | Afu3g13400 | synonymous_variant | 0.0005119 | **0.0001583** | 0.006607 |
| Chr4 | 1736561 | G | A | Afu4g06710 | synonymous_variant | **0.0002895** | **0.0001411** | 0.9852 |
| Chr4 | 1780494 | C | T | Afu4g06880 | upstream_gene_variant | **0.0002895** | **0.0001411** | 0.9852 |
| Chr4 | 1817913 | C | G | Afu4g07020 | 5_prime_UTR_variant | **0.0002379** | **0.0001198** | 0.9852 |
| Chr4 | 1840187 | A | G | Afu4g07080 | upstream_gene_variant | **0.0001657** | 0.000993 | 0.5007 |
| Chr4 | 1846648 | C | T | Afu4g07110 | upstream_gene_variant | **0.0001657** | 0.000993 | 0.5007 |
| Chr4 | 1857750 | C | T | Afu4g07140 | 5_prime_UTR_variant | **0.0001657** | 0.000993 | 0.5007 |
| Chr4 | 3134113 | A | C | Afu4g11870 | missense_variant | **0.0001782** | **0.0001607** | 0.07789 |
| Chr5 | 2400034 | T | C | Afu5g09320 | synonymous_variant | 0.02916 | 0.02272 | **0.0002492** |
| Chr5 | 2799527 | G | A | Afu5g10940 | upstream_gene_variant | **0.0003074** | **0.0002005** | 0.02931 |
| Chr5 | 3545720 | A | G | Afu5g13510 | missense_variant | **0.0004059** | 0.0002642 | 0.9872 |
| Chr6 | 1516436 | C | T | Afu6g06880 | 3_prime_UTR_variant | 0.001104 | **0.0002029** | 0.3037 |
| Chr6 | 1736813 | T | C | Afu6g07640 | synonymous_variant | **0.000272** | 0.0002558 | 0.9837 |
| Chr7 | 371671 | C | T | Afu7g01430 | upstream_gene_variant | 0.001987 | 0.0008752 | **7.96E−05** |
| Chr7 | 374641 | T | C | Afu7g01440 | missense_variant | **5.94E−05** | **3.21E−05** | 0.9782 |
| Chr7 | 409263 | C | A | Afu7g01560 | synonymous_variant | **0.0001303** | **0.0001009** | **8.56E−05** |
| Chr7 | 532236 | T | A | Afu7g01970 | missense_variant | 0.002696 | 0.001513 | **0.0002014** |

[a]Chr, chromosome; Pos, position; Ref, reference Af293 allele; Alt, alternate allele.
[b]P-val_67, 67-sample lowest 25 $P$ value ≦ 0.00044; P-val_56, 56-sample lowest 25 $P$ value ≦ 0.00021; P-val_50, 50-sample lowest 25 $P$ value ≦ 0.00035. Statistically significant values are indicated in boldface.

**Partial protein structure modeling of Afu3g13230.** We built an AlphaFold model of the portion of Afu3g13230 containing the two missense variants associated with CPE to evaluate the possible consequence these variants could have on the protein structure (see Fig. S8) (63, 64). One of variants (Gly784Arg) is predicted to occur in an extended, unstructured region of Afu3g13230 making the placement of this residue uncertain (predicted local distance difference test [pLDDT] = 41.7) (see Fig. S8). In contrast, the other variant site (Leu501Phe) is confidently placed in the model (pLDDT = 90.43) and occurs within a region of local structure (see Fig. S8A, highlighted). This structural domain

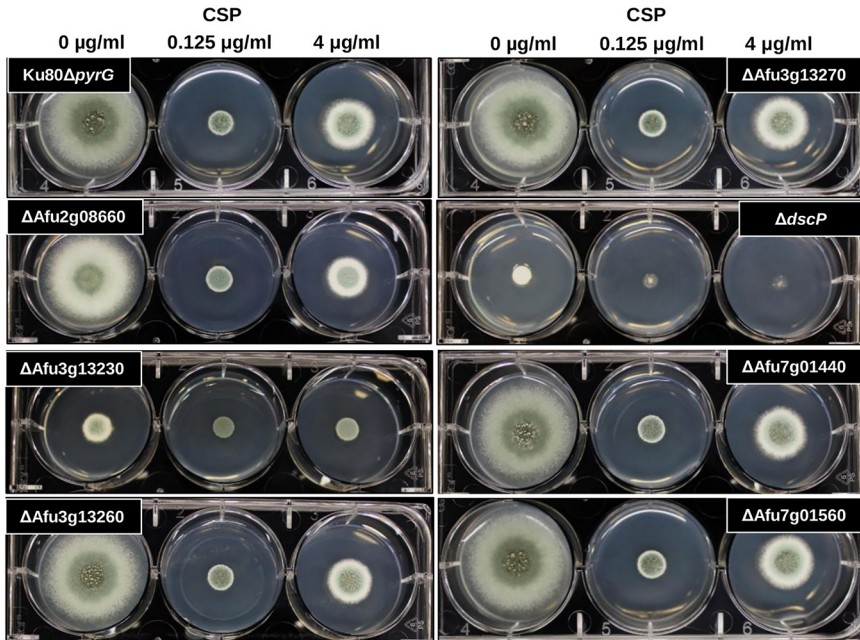

**FIG 4** Screen for presence of the caspofungin paradoxical effect among mutant strains. Ten thousand conidia were inoculated into GMM containing 0, 0.125, or 4 $\mu$g/mL of caspofungin. The culture plates were incubated at 37°C for 72 h.

comprises amino acids that are both immediately adjacent to Leu501 (AA 499 to 518) as well as more distant (amino acids 541 to 588).

Because this region of the protein is unannotated, we excised this domain from our model (see Fig. S8B) and compared it against models of known domains using DeepFRI (65). We identified 12 significant matches for the Afu3g13230 motif within the structure-based molecular function gene ontology (GO) (see Fig. S8C). All categories implicated some function related to catalytic activity acting on nucleic acids. Specifically, 8 involved nuclease activity, 2 involved hydrolase activity, and 2 involved cyclic compound binding. The portion of the domain containing Leu501 was most strongly associated with "nucleic acid binding" (GO:0003676) activity. Together, these results suggest that the large, unannotated portion of Afu3g13230 (i.e., all residues located C terminal to the already-annotated AT Hook motif) contains unannotated domains, some of which are similar in their three-dimensional structure to known functional domains.

**CEA17 gene expression of candidate genes during caspofungin exposure.** *A. fumigatus* CEA10 displays the CPE phenotype (Fig. 1A; see also Table S1). Thus, we were able to examine RNA-seq expression values of our CPE candidate genes in CEA17 (a derivative of CEA10) (66) during growth in minimal media (MM) and growth in the presence of 2 $\mu$M caspofungin (43). Of the 7 genes for which we generated knockout mutants, two were significantly differentially expressed between MM and 2 $\mu$g/mL caspofungin (see Fig. S9). The *sltB* (Afu2g08660) gene was upregulated during growth in MM (FPKM$_{MM}$ = 231.85 and FPKM$_{Caspofungin}$ = 99.95; *P* value = 1.3E−9, where FPKM represents the "fragments per kilobase million mapped reads"), while Afu3g13230 was upregulated in response to caspofungin exposure (FPKM$_{MM}$ = 13.65 and FPKM$_{Caspofungin}$ = 20; *P* = 0.004). The *dspC* gene showed a higher average expression during exposure to caspofungin, albeit this comparison was not statistically significant (FPKM$_{MM}$ = 27.57 and FPKM$_{Caspofungin}$ = 35.27; *P* = 0.11).

**STRING protein-protein interactions of candidate genes.** We investigated STRING protein-protein interactions enrichment of the protein interaction network for the genes which we experimentally knocked out (67). No interactions were present for four of the seven genes (Afu2g08660, Afu3g13230, Afu7g01440, and Afu7g01560) (Table 2). The Afu3g13260 protein-protein network showed significant KEGG enrichment for "homologous

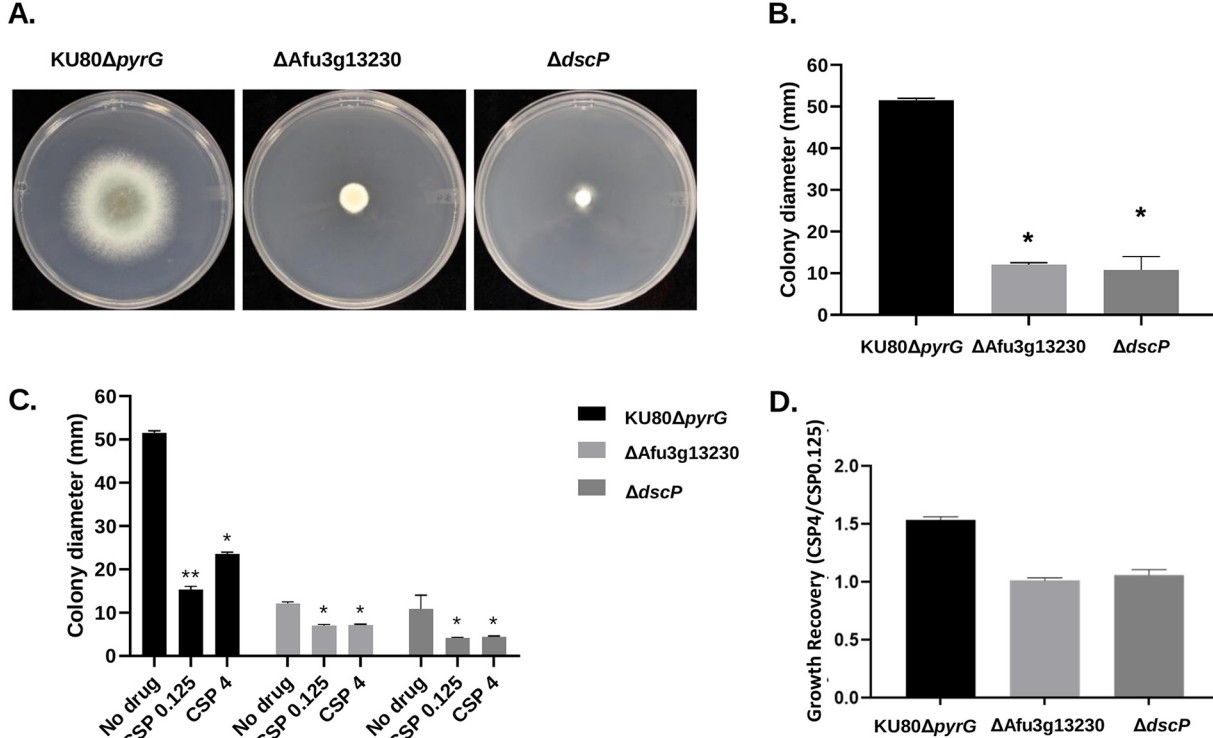

**FIG 5** Basal growth analysis and quantitative growth recovery of selected mutants during CPE. (A) Colony morphology of control and single gene deletion mutants. Ten thousand conidia were inoculated in the center of GMM agar plates and incubated at 37°C for 72 h. (B) Colony diameters were measured after 72 h of growth and compared between strains using one-way analysis of variance (ANOVA) with Tukey's multiple-comparison test (GraphPad v8.2.1). Asterisks indicate statistically significant differences ($P < 0.0001$) between deletion strains and the parental strain. (C) Ten thousand conidia were inoculated into the centers of GMM agar plates with 0.125 or 4 $\mu$g/mL of caspofungin, or without drug, followed by incubation at 37°C for 72 h. Colony diameters were measured after 72 h. Statistical analyses were carried out by two-way ANOVA with Tukey's test for multiple comparisons. Treated groups were compared between them and to the nontreated group in the case of each strain. *, $P < 0.0001$ with respect to the untreated group; **, $P < 0.0001$ between caspofungin at 0.125 and 4 $\mu$g/mL. (D) Quantification of growth recovery in the presence of different concentrations of caspofungin. Colony diameters from control and gene deletion mutant strains when grown at 4 $\mu$g/mL of caspofungin were normalized to those observed for low doses of echinocandin.

recombination," "basal transcription factors," and "nucleotide excision repair." The DgkA (Afu3g13270) protein interaction network was enriched for the KEGG terms "ether lipid metabolism," "glycerophospholipid metabolism," "glycerolipid metabolism," "endocytosis," and "biosynthesis of secondary metabolites and metabolic pathways." The DspC protein interaction network was enriched for the KEGG term "Ribosome biogenesis in eukaryotes."

## DISCUSSION

In the present study, we quantified CPE across 67 *A. fumigatus* isolates and used GWA analysis to identify alleles associated with this phenotype. A total of 61% of the isolates possessed CPE (Fig. 1). Previous smaller-scale surveys of *A. fumigatus* also revealed a relatively high frequency of CPE, though it is clear through our data—and previous data— that the magnitude of CPE is also variable (Fig. 1) (68, 69). Frequency and variability of CPE is also observed in other fungi, including *Aspergillus terreus*, *Aspergillus flavus* (68), *Candida albicans*, *Candida tropicalis*, *Candida dubliniensis*, and *Candida parapsilosis* (70–73).

In agreement with previous GWA studies in *A. fumigatus* (55, 57, 74), we demonstrate that GWA analysis paired with molecular genetics is powerful approach for identifying variants and genes that contribute to *A. fumigatus* complex traits. We conducted three independent GWA analyses and each analysis yielded a strong signal on chromosome 3, while also detecting significant signals on chromosomes 2, 4, and 7 (Fig. 3 and Table 1). The signal on chromosome 3 spanned >18 kb and 8 protein coding genes, and significantly associated SNPs were annotated as missense variants, synonymous variants, upstream variants, 5′ UTR variants, 3′ UTR variants and splice region/intron variants (Table 1). We used a CRISPR/Cas9 system to delete the chromosome 3 candidates Afu3g13230, Afu3g13260, and

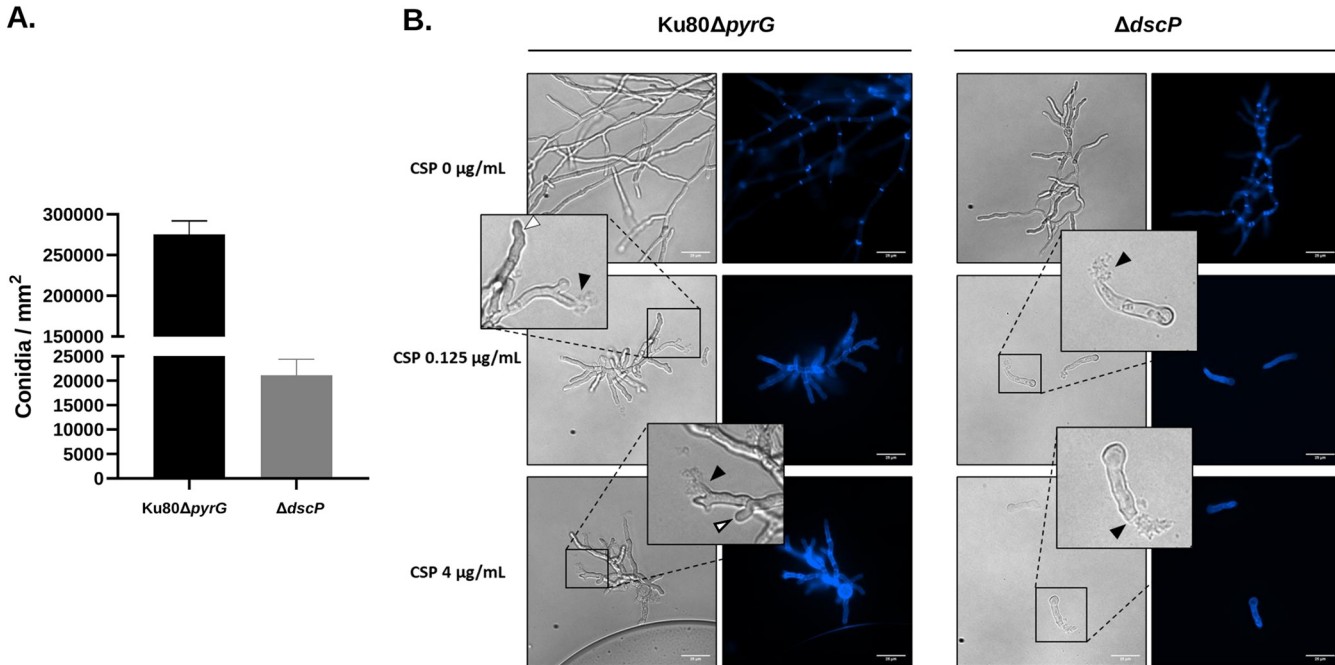

**FIG 6** DspC contributes to asexual reproduction, development, and response to stress by caspofungin in *A. fumigatus*. (A) Ten thousand conidia from the control and *dspC* deletion strains were point inoculated onto the center of GMM agar plates and incubated at 37°C for 96 h. Colony diameters were measured, and the conidia were harvested and counted. The results are expressed as means ± the standard deviations of conidia per mm² colony diameter. All experiments were performed in biological triplicate. (B) One thousand conidia were inoculated onto coverslips submerged in liquid GMM, followed by incubation at 37°C for 36 h. The coverslips were then washed and stained with 2 µg/mL of Calcofluor White (CFW). Note the presence of lysed (black arrowheads) and intact (white arrowheads) hyphal tips in the control strain grown with caspofungin, while only lysed germlings are observed in the deletion mutant.

Afu3g13270, along with candidates from the chromosome 2, 4, and 7 signals (Afu2g08660, *dscP*, Afu7g01440, and Afu7g01560) (Fig. 3).

Eight SNPs in Afu3g13300 were associated with CPE in at least one GWA analysis; however, despite several attempts, we were unable to generate a gene deletion mutant. The predicted protein encoded by Afu3g13300 is homologous to the *A. nidulans* protein NupA, which targets Nup2 to its appropriate interphase and mitotic locations (75). *A. nidulans* Δ*nupA* gene deletion mutants result in mitotic defects which include a failure of the nuclear pore complex nuclear basket-associated component Mad2 to accumulate normally in postmitotic G$_1$ nuclei (76). However, the amino acid identity between the Afu3g13300 protein and NupA is 46%, which is more divergent than the

**TABLE 2** KEGG enrichment of protein-protein network genes in CPE candidate genes

| Gene | KEGG pathway | Description | Count in network | Strength | FDR[b] |
|---|---|---|---|---|---|
| Afu2g08660 | –[a] | – | – | – | – |
| Afu3g13230 | – | – | – | – | – |
| Afu3g13260 | map03440 | Homologous recombination | 3 of 19 | 2.15 | 2.05E−06 |
|  | map03022 | Basal transcription factors | 4 of 30 | 2.08 | 7.44E−08 |
|  | map03420 | Nucleotide excision repair | 5 of 40 | 2.05 | 2.76E−09 |
| Afu3g13270 | map00565 | Ether lipid metabolism | 5 of 13 | 2.54 | 1.80E−11 |
|  | map00564 | Glycerophospholipid metabolism | 11 of 49 | 2.31 | 1.64E−24 |
|  | map00561 | Glycerolipid metabolism | 5 of 31 | 2.16 | 5.88E−10 |
|  | map04144 | Endocytosis | 3 of 67 | 1.61 | 9.76E−05 |
|  | map01110 | Biosynthesis of secondary metabolites | 10 of 367 | 1.39 | 3.25E−13 |
|  | map01100 | Metabolic pathways | 9 of 934 | 0.94 | 6.17E−08 |
| Afu4g07080 | map03008 | Ribosome biogenesis in eukaryotes | 5 of 64 | 1.85 | 1.88E−08 |
| Afu7g01440 | – | – | – | – | – |
| Afu7g01560 | – | – | – | – | – |

[a]–, No interactions found.
[b]FDR, false discovery rate.

average ortholog identify between *A. nidulans* and *A. fumigatus* (~68%) (77), suggesting the function of the Afu3g13300 protein and NupA may not be conserved. Thus, we focus our discussion mainly on Afu3g13230 and *dspC*, since deletion mutants of these genes resulted in altered CPE phenotypes.

Our analysis showed that ΔAfu3g13230 and Δ*dspC* gene deletion mutants resulted in basal reductions in growth rate, in addition to a loss of the CPE phenotype (Fig. 4 and 5; see also Fig. S7), while overexpression mutants maintained the CPE phenotype, and growth patterns were nearly identical to the wild type (see Fig. S7). Afu3g13230 contains domains with predicted DNA-binding activity. Interestingly, this gene was also upregulated in CEA17 (a CPE[+] strain) during exposure to caspofungin (43) (see Fig. S9). We identified two variants within the coding region of this gene that were associated with CPE, both of which encoded missense variants (Gly784Arg and Leu501Phe) (see Fig. S5). The Gly>Arg variant was annotated as a moderate impact mutation by SnpEff, as glycine is small, uncharged and nonpolar, while arginine is positively charged and polar. The Leu>Phe variant is also annotated as a moderate impact mutation by SnpEff, since leucine is aliphatic in comparison to phenylalanine which is aromatic. Interestingly, a model of the Afu3g13230 protein region containing these variant sites revealed that the Leu>Phe site resides in a region of local structure (see Fig. S8), suggesting that missense variants have the potential to destabilize and disrupt functional structural features, potentially impacting phenotype.

*dspC* encodes a protein with predicted tyrosine phosphatase activity. The *Saccharomyces cerevisiae* ortholog, *Yvh1*, encodes a protein that is required for release of the nucleolar ribosome-like protein Mrt4 from the pre-60S ribosomal particles and is thus required for assembly of the 60S ribosomal subunit (78). The SNP associated with CPE was present ~3 kb upstream of the *dspC* start codon. Interestingly, a previous effort to knockout *dspC* resulted in lethality (79). However, here, our Δ*dspC* gene deletion mutant resulted in reduced growth rate compared to the WT (Fig. 4 and 5; see also Fig. S7 and S8). Similarly, growth rate analysis of the Δ*yvh1* mutant in *Candida albicans* also revealed inhibition of growth which was attributed to delay in nuclear division and septum formation, as well as a decrease in virulence in a mouse model (80). Knockout of the *A. flavus* ortholog (*AFLA_112770*) resulted in defects in conidiation and sclerotia development, in addition to aflatoxin production, indicating the role of *Yvh1* in development (81). In agreement with these results, the *dscP* deletion mutant also shows deficient conidiation capacity. We observed that the WT produced 13 times more conidia than the gene deletion mutant (Fig. 6). Δ*hspA-dscP* overexpression mutants maintained the CPE phenotype (see Fig. S7B), and *dscP* is expressed in the absence or presence of caspofungin (see Fig. S9). For both Afu3g13230 and *dscP*, gene inactivation, as modeled by our gene deletion mutants, likely does not represent the mechanisms by which variants influence CPE. However, the observation that (i) these genes are expressed in the presence of caspofungin in a CPE[+] isolate (43), (ii) CPE is lost in gene deletion mutants, and (iii) CPE is maintained in overexpression mutants, collectively suggest that the expression of Afu3g13230 and *dscP* are necessary for CPE and changes to protein structure or different transcription level alterations could influence CPE.

We used STRING to explore protein-protein interactions with our GWA candidate genes (67) and the protein-protein interaction network of DspC showed an enrichment of genes in the "ribosome biogenesis in eukaryotes" KEGG pathway (Table 2). Previous transcriptome profiling of *A. fumigatus* CEA17 (CPE[+]) and Δ*fhdA* mutant (CPE[−]) revealed that the WT CPE[+] isolate displayed downregulation of genes involved in rRNA processing and ribosome biogenesis in the presence of caspofungin (43). Further, the CPE[−] mutant showed an upregulation of ribosome biogenesis genes in the presence of caspofungin in comparison to the WT CPE[+] isolate (43). Transcriptomic and proteomic analysis of *Candida albicans* and *Candida auris* in the presence of caspofungin also showed an enrichment of differentially expressed genes and proteins involved in ribosomal function (82–84). The general interaction between caspofungin and ribosome biogenesis could represent a metabolic reshuffling in response to caspofungin, and *dspC* is an interesting candidate gene because of its close association with ribosome biogenesis (78). Our work opens new avenues for

the characterization of genes involved in caspofungin resistance and tolerance. Future work should focus on the introduction of point mutations in these selected genes and investigation of their phenotypes and should use long-read DNA sequencing to characterize structural variants that may be linked to the variants associated with CPE but undetectable via short-read sequencing.

## MATERIALS AND METHODS

**Quantifying CPE.** Growth rate was measured for the 67 isolates on MM without or with 0.125, 0.25, 0.5, 1, and 8 $\mu$g/mL caspofungin. Colony diameter was measured after growth at 37°C for 96 h. All experiments were performed in duplicate, and average values were used for subsequent analysis. To measure the paradoxical effect we calculated the recovery rate, as follows:

$$\text{recovery rate} = \frac{\text{colony diameter}\left[8\,\mu\text{g/ml}\right] - \text{minimum colony diameter}}{\text{colony diameter}\left[0\,\mu\text{g/ml}\right] - \text{minimum colony diameter}}$$

We considered isolates with recovery rates of $\geq$0.1 as possessing the CPE phenotype (CPE$^+$) and isolates with recovery rates $<$ 0.1 as not possessing the CPE phenotype (CPE$^-$).

**A. fumigatus whole-genome Illumina data.** Whole-genome paired-end Illumina resequencing data were downloaded from the NCBI Sequence Read Archive (SRA) using the following run accession numbers: A1163 (SRR068950), Af293 (SRR068952), AF72 (SRR617721), AfS35 (DRR146814), Afu-AF10 (SRR334209), ATCC 204305 (SRR7418943), ATCC 46645 (SRR7418935), CEA10 (SRR7418934), CM2141 (SRR7418947), CM237 (SRR7418942), CM3248 (SRR7418945), CM5419 (SRR7418944), CM5757 (SRR7418949), CM6126 (SRR7418937), CM6458 (SRR7418936), CM7510 (SRR7418939), CM7555 (SRR7418938), CNM-CM2495 (SRR7418930), CNM-CM2730 (SRR7418924), CNM-CM2733 (SRR7418923), CNM-CM3249 (SRR7418926), CNM-CM3262 (SRR7418922), CNM-CM3720 (SRR7418927), CNM-CM4602 (SRR7418928), CNM-CM4946 (SRR7418948), CNM-CM7560 (SRR7418925), CNM-CM7632 (SRR7418941), CNM-CM8057 (SRR10592633), CNM-CM8686 (SRR10592630), CNM-CM8689 (SRR10592629), CNM-CM8714 (SRR10592632), CNM-CM8812 (SRR10592631), F12041 (SRR617723), F12636 (SRR617725), F13535 (SRR617726), F13952 (SRR617728), F14946 (SRR159252), F17764 (SRR617745), F7763 (SRR617744), IF1SW-F4 (SRR4002444), IFM55369 (DRX013572), IFM58026 (DRX015829), IFM58029 (DRX015830), IFM59056 (DRX013573), IFM59073 (DRX013577), IFM59359 (DRX013574), IFM59361 (DRX013575), IFM59365 (DRX015832), IFM59777 (DRX015833), IFM60514 (DRX013576), IFM61118 (DRX015834), IFM61407 (DRX013578), IFM61578 (DRX015835), IFM61610 (DRX013579), IFM62516 (DRX015837), ISSFT-021 (SRR4002443), MO54056 (SRR5676587), MO68507 (SRR5676586), MO69250 (SRR5676589), MO78722 (SRR5676591), MO79587 (SRR5676590), MO89263 (SRR5676593), MO91298 (SRR5676592), TP-12 (SRR7418940), and TP32 (SRR7418946). The sources of all samples are presented in Table S1 (85–94). Genomic DNA and Illumina whole-genome sequencing of samples 17993925 and 200-89320 were carried out as previously described (11, 57). Raw sequencing data for 17993925 and 200-89320 were deposited to the NCBI SRA under accession numbers SRR16287627 and SRR16287628, respectively.

**Variant calling.** Variant calling was conducted as previously described (57). Briefly, raw Illumina reads were adapter and quality trimmed using trim_galore v0.4.2 with the following parameters: "–stringency 1," "-q 30," and "–length 50." Paired-end trimmed reads were then mapped against the *A. fumigatus* Af293 reference genome (95) using BWA-MEM v0.7.15 (96). SAM files were converted to sorted BAM format with SAMtools 1.4.1 (97). SNP genotyping was conducted with GATK v4.0.6.0 using the best practice pipeline for "Germline short variant discovery" (98, 99). "HaplotypeCaller" was used to call short variants (SNPs and indels) in each sample. Next, the "GenotypeGVCFs" function was used to generate a joint-called variant file using combined g.vcf file. To reduce false-positive variant calling, "VariantFiltration" was used to apply "hard filtering" with the following parameters: "QD $<$ 25.0 $\parallel$ FS $>$ 5.0 $\parallel$ MQ $<$ 55.0 $\parallel$ MQRankSum $<$ $-$0.5 $\parallel$ ReadPosRankSum $<$ $-$2.0 $\parallel$ SOR $>$ 2.5." 181,309 SNP sites were predicted after filtering.

**Population structure analysis of A. fumigatus isolates.** Using the entire matrix of 181,309 SNPs, we conducted principal-component analysis (PCA) in Tassel v5 (100) to examine the relationship between isolates. We also conducted phylogenetic network analysis to examine population structure. For this analysis, we used vcftools v0.1.14 to space SNPs by a minimum of 4 kb in an effort to minimize the effect of linkage (101). The resulting data set consisted of an SNP alignment of 6,492 SNP sites distributed across the genome. Phylogenetic network analysis was conducted in splitstree v4.16.1 using the neighbornet method and 1,000 bootstrap replicates (102).

**Genome-wide association analysis of CPE.** We conducted genome-wide association (GWA) analysis to identify genetic variants associated with CPE using PLINK v1.9 (59). We conducted three independent GWA analyses. First, GWA was applied on the entire set of 67 samples, treating the data as quantitative, and correcting for population structure using four principal components (PCs). In our next GWA analyses, we excluded 11 closely related isolates that belonged to the most distant population (population A). In this GWA, we analyzed 56 samples treating the data as quantitative, and correcting for population structure using 4 PCs. In the third GWA, analysis was conducted using 50 samples, treating the data at binary (CPE$^+$ or CPE$^-$) and using 1 PC for population structure correction. The 25 samples with the largest recovery rate, and 25 samples with the lowest recovery rate were used for analysis. Quantile-quantile (Q-Q) plots were generated using the R package "qqman" (103) in order to evaluate potential *P*-value inflation and population structure overcorrection. The putative functional effects of candidate SNPs were predicted using SnpEff v4.3t (104) with the *A. fumigatus* Af293 reference genome annotation.

**Gene deletion and overexpression strain generation.** We functionally validated a subset of our candidate genes by deleting them using a CRISPR/Cas9 approach (62, 105). Candidates for gene deletion were selected as follows: first, we required that SNPs had a significant association in the 67 sample GWA analysis. Next, we prioritized candidate genes based on the associated SNP being identified in more than one GWA analysis (e.g., Afu3g13230, Afu3g13260, Afu3g13270, and Afu7g01560, which were detected in all three GWA analyses and Afu7g01440 detected in the 67-sample and 56-sample GWA analyses) or based on their functional annotation (e.g., kinases and phosphatases that may be involved in responding to external stimuli such as Afu3g13270 and Afu4g07080 [*dscP*]). We also attempted to generate gene deletion mutants for Afu3g13300 (significant in all GWA analyses) and Afu2g08670 (significant in the 67-sample GWA), but we did not obtain transformants after several attempts, suggesting these genes may be essential.

To determine whether the candidate genes were important for CPE or for caspofungin susceptibility in *A. fumigatus*, each complete open reading frame (ORF) was deleted in the uracil auxotrophic KU80Δ*pyrG* (CEA17) genetic background (106), by replacing each with the *Aspergillus parasiticus pyrG* gene using a CRISPR-Cas9 gene editing technique (105). Briefly, 20-bp protospacer sequences were selected immediately upstream and downstream of the start and stop codons, respectively, for each gene (Table 3). After screening against potential off-site targeting, each protospacer sequence was then utilized to generate a guide-RNA through a commercial vendor. The gRNAs were employed, in conjunction with commercially available Cas9 enzyme, to generate ribonucleotide-protein (RNP) complexes to mediate double-strand DNA breaks for gene replacement, as previously described (105). A repair template containing the *A. parasiticus pyrG* gene, with ~600 bp of endogenous promoter and ~1,500 bp of terminator region, was amplified from plasmid pJW24 (107) to contain 35-bp microhomology regions directed toward the targeted genes. Protoplasts were generated and transformed, as previously described (105, 108), and plated onto osmotically stabilized minimal medium. Plates were incubated overnight at room temperature and then transferred to 37°C until colonies were observed. Potential transformant colonies were genotypically screened using multiple PCRs to ensure a correct integration at the target site.

Because Afu3g13230 and *dscP* gene deletion mutants resulted in the loss of CPE, we generated overexpression mutants of both genes to assess the phenotype. To induce overexpression of Afu3g13230 and *dscP*, the endogenous promoters were replaced by the *A. fumigatus* heat shock protein A (*hspA*) promoter in the Δ*akuB-pyrG*$^+$ genetic background (109). Briefly, a repair template containing a hygromycin resistance cassette followed by the *hspA* promoter was amplified from plasmid pJMR2 (62) and contained microhomology regions of ~40 bp. The gRNAs 3g13230 5' and 4g07080 gRNA 5' (Table 3) were used in these cases. Transformation and screening were performed as described for the gene deletion mutants.

**CPE quantification in mutants.** To quantify colony development under caspofungin stress, five $\mu$L containing $10^4$ conidia were spot inoculated onto the centers of glucose minimal media (GMM) plates containing 0.125 or 4 $\mu$g/mL of caspofungin (110). GMM agar plates containing no drug were inoculated and used as nontreatment controls. The auxotrophic KU80Δ*pyrG* strain was used as the control parental strain and, consequently, uridine and uracil were added to the culture medium to allow growth. The culture plates were incubated at 37°C for 72 h, and the colony diameters were measured every 24 h. The experiment was performed in biological triplicates, and data are provided as means $\pm$ the standard deviations.

**Quantification of conidium production in the Δ*dscP* mutant.** To evaluate the capacity of generating conidia in the *dscP* gene deletion mutant, 5-$\mu$L suspensions containing $2 \times 10^4$ conidia from the control and mutant strains were point inoculated in the center of GMM plates and allowed to grow for 96 h at 37°C. After this, the colony diameters were measured, and conidia were harvested from each plate and counted using a hemocytometer. Conidial abundances from three biological replicates were compared using a *t* test (GraphPad v9.2), and the results are expressed as conidia per mm$^2$ of colony area.

**CEA17 gene expression of candidate genes in absence and presence of caspofungin.** We examined gene expression of *A. fumigatus* CEA17, which is CPE$^+$, by accessing RNA-seq-based gene expression quantification values from a previous study (43). Gene expression values (fragments per kilobase million mapped reads [FPKM]) and *P* values were generated by DESeq2 (111).

**Protein modeling and structure-based annotation of Afu3g13230.** To find a protein model for Afu3g13230, we first performed a sequence-based similarity search (NCBI BLASTP) of the PDB, AlphaFold DB, and UniProtKB PDB. No existing structural models were found that matched our query with an identity >30%. Next, we constructed an AlphaFold2 model using the canonical Afu3g13230 peptide sequence. Because Afu3g13230 is an extremely large protein, we focused on the portion of the protein that contained the two CPE$^-$ variants (amino acids 501 to 784), supplementing this core sequence with that of the 100 amino acids flaking on either side. The resulting peptide sequence of 483 amino acids was then processed through AlphaFold2, using ColabFold (65). Four models of Afu3g13230 were generated, and the best model (based upon overall model confidence) was used in the analysis here.

To classify the subdomain identified in the AlphaFold2 model, we extracted the domain from the larger Afu3g13230 structure (see Fig. S8) and submitted for evaluation by DeepFRI (65). DeepFRI uses graph convolutional networks with language model features to predict protein function from a structural model. DeepFRI outputs scored GO terms along with per-residue salience scores. DeepFRI scores of >0.5 are considered significant.

**Protein-protein interactions and KEGG enrichment of candidate genes.** We investigated protein-protein interactions and KEGG term enrichment for the seven candidate genes for which we experimentally generated gene deletion mutants using STRING v11.0 (67).

**TABLE 3** Primers and crRNAs used in this study

| Primer | Sequence (5′–3′)[a] |
|---|---|
| pyrG Fw 2g08660 | TTCCCCCTAATCACTGCACCCTTTCCCCGGACTCTGCACGgaattctcatgtttgacagc |
| pyrG Rv 2g08660 | GTGATATAGAGACGAGGACAAATATGCTAGAAGCTTTATGggatccacaggacgggtgtgg |
| 2g08660 Fw Scr | GACTCAGACCCGCTTCGC |
| 2g08660 Rv Scr | CCACGCACGATATCCATGACC |
| pyrG Fw 2g08670 | TGCGTGGGAATTGTATAATTTATAATAAAACTGCAAGCATgaattctcatgtttgacagc |
| pyrG Rv 2g08670 | ACGGGGCGGTTCTATGTTTCATTCTATTTTCATGGTGAGCggatccacaggacgggtgtgg |
| 2g08670 Fw Scr | ACTTTCGCGTCCTCACCTC |
| 2g08670 Rv Scr | CCTAACGCACCCGCCTTG |
| pyrG Fw 3g13230 | CGGCACGATCGCCCCTCTTCATGAACCGCATTGTTTGTTGgaattctcatgtttgacagc |
| pyrG Rv 3g13230 | GATCTCAAAGTCGAGGGTGATGGATGAACGGTCATCGACGggatccacaggacgggtgtgg |
| 3g13230 Fw Scr | GGTGCGTTGGCAATACAACC |
| 3g13230 Rv Scr | ACCGGATCCGTAATAGTCCG |
| pyrG Fw 3g13260 | TCTGTTTTTTAGGCTTTCACTTGTGGTCTCTGGTCTGTCTgaattctcatgtttgacagc |
| pyrG Rv 3g13260 | TAACAAATCAACTTTGAAGTCGGCAAACATCAACTCAAGTggatccacaggacgggtgtgg |
| 3g13260 Fw Scr | GCCTGCTTCACAGTAGTCGAG |
| 3g13260 Rv Scr | CATAGACGCCAGAACACCGC |
| pyrG Fw 3g13270 | ACCTCTGTCCGGGCTGATACCGGCTCCGGGTCTTCCGCCTgaattctcatgtttgacagc |
| pyrG Rv 3g13270 | AGCGCGTCAGAGAGAAATTGACAGGCCCGACCTCGGCATTggatccacaggacgggtgtgg |
| 3g13270 Fw Scr | TGCGTCGTTTGCAGCTGG |
| 3g13270 Rv Scr | CCATTGCGGATGTCCCATCG |
| pyrG Fw 3g13300 | GCCTTGGTGCCGTTCTTTCAGTCGGTTGTTGTTCTGCTTTgaattctcatgtttgacagc |
| pyrG Rv 3g13300 | TATCCAGAGCCTTTCGTCACCCTGATGACATGACTTGAGggatccacaggacgggtgtgg |
| 3g13300 Fw Scr | CGTCTCAAATCACGTCGCGG |
| 3g13300 Rv Scr | ACAGGCTATGCTCAGTAATCGG |
| pyrG Fw 4g07080 | TAATTTTGTCCTTCGCTGATCGGCTATAACGTGGCATCAGgaattctcatgtttgacagc |
| pyrG Rv 4g07080 | AGAGTTCATCCCAGACGATGGGTAGATTAGATTCTGAGGAggatccacaggacgggtgtgg |
| 4g07080 Fw Scr | ACCTCCCCTTAACTCGTCACC |
| 4g07080 Rv Scr | CGGTCTGCTAACGTAAGCCG |
| pyrG Fw 7g01440 | GCAGCACGCATCTCAGCATTAGTACTCAACAGATGGAAACgaattctcatgtttgacagc |
| pyrG Rv 7g01440 | TATAGCCTTGGGAAAGGGTAGATGGTAGAGGATACCGTTTggatccacaggacgggtgtgg |
| 7g01440 Fw Scr | TTCAGCAGTGTCAGCACTCGG |
| 7g01440 Rv Scr | AGCCATCGACGTTCACGC |
| pyrG Fw 7g01560 | TCTATAAGTTGTTCTAAGCCCATAACGCACCATCATGACCgaattctcatgtttgacagc |
| pyrG Rv 7g01560 | TGGTCCAAGATCAATATAAGACAGCATATCGTTAATGAAGggatccacaggacgggtgtgg |
| 7g01560 Fw Scr | GCAACGTCTGCCGCGATG |
| 7g01560 Rv Scr | TCAGTTGGTGGCACAGTCC |
| pyrG Fw 8g06360 | TTCATGGTATGATCATGCTGACTTTGCTCGAACTTAATGAgaattctcatgtttgacagc |
| pyrG Rv 8g06360 | AAGAACGGCTGCTCTTGTATGGTGTGCATCTACAATGCTTggatccacaggacgggtgtgg |
| 8g06360 Fw Scr | GCCTGGCAGATGCAAAGGC |
| 8g06360 Rv Scr | AGTACCCTCGGCAAGCGC |
| Ap pyrG Fw scr | gcccttgcagagaagcac |
| Ap pyrG Rv scr | cagcataaattccacgaccagc |
| Hyg Fw 3g13230 | GCGCGACGCGATTCGCGGCACGATCGCCCCTCTTCATGAACCGagcttgcatgcctgcaggtcg |
| hspA Rv 3g13230 | CCAAGAATGTCTGGAGAAGAGCTGGAACTTTGAGAGAACATtgtgaagaagtgaggaggg |
| Hyg Fw 4g07080 | GAAACTATACTCATAATTTTGTCCTTCGCTGATCGGCTATAAagcttgcatgcctgcaggtcg |
| hspA Rv 4g07080 | CTCCGATGTAGATATCGTGTCCTGGGATCTTGTTCATAGCCATtgtgaagaagtgaggaggg |
| 2g08660 5 gRNA | AGATGTCCGTTCTCCGTCAT |
| 2g08660 3 gRNA | TCGCAGCCTAGAAAGACAGC |
| 2g08670 5 gRNA | TAATAAAACTGCAAGCATCA |
| 2g08670 3 gRNA | GGAGGCCGTGCTCAAGTACC |
| 3g13230 5 gRNA | GAACCGCATTGTTTGTTGCA |
| 3g13230 3 gRNA | GAGGAGCCGCTCTCACGGCG |
| 3g13260 5 gRNA | GTGGTCTCTGGTCTGTCTCA |
| 3g13260 3 gRNA | CATCCCCGTGTAAGCATTT |
| 3g13270 5 gRNA | GCTCCGGGTCTTCCGCCTTT |
| 3g13270 3 gRNA | TAGACATATACCTCTTTCCT |
| 3g13300 5 gRNA | GTCTACAGTCAAGATGCGTA |
| 3g13300 3 gRNA | ATTTACGAAAGCTCAAGCCT |
| 4g07080 5 gRNA | GCTATAACGTGGCATCAGCA |
| 4g07080 3 gRNA | GGCCGCGGCAATCTTTGACG |
| 7g01440 5 gRNA | TACTCAACAGATGGAAACAT |
| 7g01440 3 gRNA | AAGAATGGTTAATCATCATT |

**TABLE 3** (Continued)

| Primer | Sequence (5′–3′)[a] |
|---|---|
| 7g01560 5 gRNA | TAACGCACCATCATGACCCG |
| 7g01560 3 gRNA | AGAGAACTAGCGGAGGATTC |
| 8g06360 5 gRNA | AAGTGCTAAGAATGCATTTA |
| 8g06360 3 gRNA | ACAAGCATTGTAGATGCACA |

[a]Sequences for the amplification of the *A. parasiticus pyrG* cassette are indicated in lowercase, while the microhomology arms for each gene are indicated in uppercase.

**Data availability.** Raw whole-genome Illumina data for the 67 isolates are available through the NCBI SRA through the accession numbers listed in Materials and Methods and in Table S1 in the supplemental material.

## SUPPLEMENTAL MATERIAL

Supplemental material is available online only.

**SUPPLEMENTAL FILE 1**, PDF file, 3.7 MB.

## ACKNOWLEDGMENTS

This research was supported by grant R21AI137485 from the National Institutes of Health and National Institutes of Allergy and Infectious Diseases (NIAID) to J.G.G., which supports J.G.G. and S.Z. This study was supported in part by NIH/NIAID awards R21AI142509 and R01AI158442 to J.R.F. We also acknowledge São Paulo Research Foundation (FAPESP) grants 2021/04977-5 (G.H.G.), 2016/21392-2 (L.P.S.), and 2017/07536-4 (A.C.C.) and Conselho Nacional de Desenvolvimento Científico e Tecnológico (CNPq; G.H.G.), both from Brazil, for financial support. Computational analysis was conducted on the Massachusetts Green High Performance Computing Center.

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
