## [Reviewer comments · Microbiology Spectrum]

Microbiology Spectrum

Genomic and molecular identification of genes contributing to the caspofungin paradoxical effect in *Aspergillus fumigatus*

Shu Zhao, Adela Martin-Vicente, Ana Cristina Colabardini, Lilian Pereira Silva, David Rinker, Jarrod Fortwendel, Gustavo Goldman, and John Gibbons

Corresponding Author(s): John Gibbons, University of Massachusetts Amherst

Review Timeline:

Submission Date:	February 9, 2022
Editorial Decision:	March 10, 2022
Revision Received:	August 4, 2022
Accepted:	August 17, 2022

Editor: Rhys Farrer

Reviewer(s): Disclosure of reviewer identity is with reference to reviewer comments included in decision letter(s). The following individuals involved in review of your submission have agreed to reveal their identity: Teppei Arai (Reviewer #2)

Transaction Report:

DOI: <https://doi.org/10.1128/spectrum.00519-22>

March 10, 2022

Dr. John G Gibbons
University of Massachusetts Amherst
Food Science
240 Chenoweth Lab
102 Holdsworth Way
Amherst, MA 01003

Re: Spectrum00519-22 (Genomic and molecular identification of genes contributing to the caspofungin paradoxical effect in *Aspergillus fumigatus*)

Dear Dr. John G Gibbons:

Thank you for submitting your manuscript to Microbiology Spectrum.

Your manuscript has been read by two reviewers.

While the reviewers highlight several positive aspects about your manuscript, reviewer #1 in particular highlights several areas that are lacking - especially the apparent lack of reconstituted controls for the gene deletions. Without those controls, it would not be possible to determine if knockout phenotypic changes are a result of off targets. Given the reviewers recommendations, i will also recommend modifications - although this issue, and the other major and minor issues flagged by the reviewers will need to be sufficiently addressed to be reconsidered.

Link Not Available

Sincerely,

Rhys Farrer

Journals Department
Reviewer comments:

Reviewer #1 (Comments for the Author):

I apologize for having taken overly long for this review!

In the manuscript presented by Zhao et al, the authors describe a genome-wide association study on the caspofungin paradoxical effect in *Aspergillus fumigatus*.

They investigated 67 isolates, which were partially redundant (clonal), but mostly well distributed across the phylogenetic tree of *A. fumigatus*.

The authors then derive a list of candidate genes by aligning SNPs derived from publicly available genome sequences of the isolates included in their study along phenotypic laboratory data. From this list, they identify a cluster of SNPs on Chr3 potentially involved in CPE, and in several other genes across the genome.

Based on the list, seven genes were chosen for further genetic analysis by constructing gene deletion strains. Deletions of two of the genes of the cluster were found to impact on CPE (Afu3g13230 and Δ dspC).

I find the study interesting to read, and definitely an important endeavor, given that CPE may have clinical impact. To the best of my knowledge, the two aforementioned genes have not previously been implicated with this phenotype. Non-synonymous SNPs in these genes potentially impacting on protein function are discussed along bioinformatics analyses. The manuscript is generally well written, although I have few comments on the rationales and methods as listed below.

Methodology:

The reference genome used for deducing SNPs is Af293, which is a strains that is placed outside the major phylogenetic body. AF1163 would be a more appropriate reference.

Table 2 lists SNPs leading to non-synonymous changes in orfs along with synonymous and those outside coding regions. I am not entirely sure how the gene selection process was done for creating the mutants, this should be described better. Were only non-synonymous SNPs used for selection?

I am missing a more wholistic discussion of the 18kb gene cluster on Chr3. A map of that particular locus would be helpful, including possible functional explanations and SNPs observed, and regions deleted.

Most importantly, the genetic approach is lacking reconstituted controls for the gene deletions, and does not compensate for possible off-target effects through CRISPR. Here, genetic replicates (figure S6) are the only control. I am not sure if this is up to standards, please discuss this in a potential rebuttal.

The study also relies on the assumption that CPE is a gain-of-function phenotype, it might just as well be the opposite, so that continuous (over)expression might actually show the expected phenotype, or its loss (depending on the perspective and strain used). SNPs in potential regulatory regions (transcriptional upregulation of Afu3g13230 in CEA17!) are completely ignored. Also the transcriptional measurements are not discussed further, but might help the reader.

Since the authors are using CRISPR, it would have been a more informative approach to test these individual SNPs directly, instead of modifying the entire region by deleting larger chunks (aka genes).

In summary, as far as the wet-lab experiments in this study go, they are sound and the implication of the two genes in CPE is derived from solid data. However, there might be more potential in the data and it the study could benefit from a more in-depth analysis of individual SNPs in both genetic directions.

I have no previous expertise with GWA studies, and can therefore not judge the bioinformatics approach used to determine SNPs and their ranking.

Text and formatting issues:

Results obtained for gene deletions outside chr3 are omitted in the discussion, please include at least one sentence

Lines 124-134 could be removed: there is no doubt GWA is a useful method in fungi

Lines 138ff: create a better list of concentrations, e.g. "without and with 0.125, 0.25, 0.5, 1, and 8 mg/l, respectively. Or similar Paragraph with accession numbers: please integrate numbers into table S1. Use the space to briefly describe the sequencing technology used in the references 11 and 55 instead.

Line 214 and throughout text: replace "knockout" by "gene deletion" or "gene inactivation" as appropriate.

Line 262: remove "(see methods)"

Figure 3 gave me a lot of trouble in the pdf because the dots are all individual objects. Please make sure that another figure format is chosen (e.g. tiff)

Figure4. move label for dAfu3g13270 to right. Add human-readable gene names to make interpretation easier, where available.

Figure 5B/C and 6A: in the pdf sometimes there is a rho instead of a delta preceding the gene names (where I think a delta should be). Please change, or elaborate (perhaps I am just not familiar with that nomenclature)

Table S1, in addition to the adding accession numbers, please rename the last column to something that makes clear this is not the reference for the phenotypic data, but only for the strain source. Remove concentrations from table legend, as they are also in the table.

Reviewer #2 (Comments for the Author):

This work focused on the "Caspofungin Paradoxical Effect" and searched for causative genes by genome-wide association (GWA) analysis. And, two genes were shown to contribute to CPE. The methods and conclusions are scientifically sound. I would like to make a few comments, which I hope you will find useful.

Comments

1. Introduction section Line 87 "while azoles disrupt ergosterol biosynthesis resulting in toxic sterols (31)."
This statement is misleading. This is one hypothesis, so please rephrase it.

2. Results, CRISPR/Cas9 gene deletion of candidate genes, section Line 321-322

"~ a putative diacylglycerol kinase (detected in all GWA), DspC, a predicted tyrosine phosphatase (87) (detected in 67-sample GWA), Afu7g01440,~" ;

Please correct the following statement.

"~ a putative diacylglycerol kinase (detected in all GWA), Afu4g07080, which encodes DspC, a predicted tyrosine phosphatase (87) (detected in 67-sample GWA),~"

And, it would be helpful to add a table with gene ID, gene name, and protein to the supplemental.

3. Two names are used for one mutant like Δ dscP and Δ 4g7080. Please unify either of them.

4. Discussion section Line 421 and 424;

There are two designations, WT CPE+ isolate and CPE+ WT isolate. Do these refer to the same?

5. The data presented in Table S1 shows that the strains that have a Recovery Rate of 0 are growing well. This is different from the phenotype observed for Afu3g13230 or dspC deficient strains. It would be better to discuss this.

6. For the two genes you focused on in this work, have you examined the gene expression levels et.al. in the CPE+ isolate shown in Figure 1? There is little discussion of the contribution of the two genes to CPE in these isolates.

Staff Comments:

Preparing Revision Guidelines

Please return the manuscript within 60 days; if you cannot complete the modification within this time period, please contact me. If you do not wish to modify the manuscript and prefer to submit it to another journal, please notify me of your decision immediately so that the manuscript may be formally withdrawn from consideration by Microbiology Spectrum.

Corresponding authors may join or renew ASM membership to obtain discounts on publication fees. Need to upgrade your

membership level? Please contact Customer Service at Service@asmusa.org.

Reviewer #1 (Comments for the Author):

I apologize for having taken overly long for this review!

- In the manuscript presented by Zhao et al, the authors describe a genome-wide association study on the caspofungin paradoxical effect in *Aspergillus fumigatus*. They investigated 67 isolates, which were partially redundant (clonal), but mostly well distributed across the phylogenetic tree of *A. fumigatus*. The authors then derive a list of candidate genes by aligning SNPs derived from publicly available genome sequences of the isolates included in their study along phenotypic laboratory data. From this list, they identify a cluster of SNPs on Chr3 potentially involved in CPE, and in several other genes across the genome. Based on the list, seven genes were chosen for further genetic analysis by constructing gene deletion strains. Deletions of two of the genes of the cluster were found to impact on CPE (Afu3g13230 and Δ dspC).
- I find the study interesting to read, and definitely an important endeavor, given that CPE may have clinical impact. To the best of my knowledge, the two aforementioned genes have not previously been implicated with this phenotype. Non-synonymous SNPs in these genes potentially impacting on protein function are discussed along bioinformatics analyses. The manuscript is generally well written, although I have few comments on the rationales and methods as listed below.

Methodology:

- The reference genome used for deducing SNPs is Af293, which is a strains that is placed outside the major phylogenetic body. AF1163 would be a more appropriate reference.

We thank the review for raising this important concern, as the choice of a reference genome can bias results. We chose Af293 as the reference genome because it is, by far, the best annotated strain of *A. fumigatus*. While we agree that Af293 is in outside of the major phylogenetic body (Figure S2), this strain is still part of population D (the orange population in Figure 2 and Figure S2), of which A1163 is also a member. To investigate whether our choice of reference genome could have impacted our results, we conducted several analyses. First, we used orthofinder to identify the number of orthogroups between Af293 and A1163. Out of 19,552 protein coding genes in the two strains, only 381 genes were not assigned to orthogroups. This high level of homology suggests that little information is lost when choosing between Af293 or A1163 as a reference genome. Additionally, both isolates display CPE, and are thus likely to share variants underlying this phenotype.

- Table 2 lists SNPs leading to non-synonymous changes in orfs along with synonymous and those outside coding regions. I am not entirely sure how the gene selection process was done for creating the mutants, this should be described better. Were only non-synonymous SNPs used for selection?

We have added text describing our process of selection genes for generating deletion mutants from the GWA data. First, we required that SNPs have a

significant association in the 67 sample GWA analysis. We prioritized candidate genes for gene deletion based on the associated SNP being identified in more than one GWA analysis (e.g. Afu3g13230, Afu3g13260, Afu3g13270 and Afu7g01560 which were detected in all three GWA analyses and Afu7g01440 detected in the 67-sample and 56-sample GWA analyses) or based on their functional annotation (e.g. kinases and phosphatases that may be involved in responding to external stimuli such as Afu3g13270 and Afu4g07080 (*dscP*)).

- I am missing a more wholistic discussion of the 18kb gene cluster on Chr3. A map of that particular locus would be helpful, including possible functional explanations and SNPs observed, and regions deleted.

Thanks for this excellent suggestion. We created a new supplemental figure showing the schematic of the chromosome 3 region and the SNPs in this region associated with CPE. We have also added some detail of Afu3g13300 to the discussion, as this gene contained 8 SNPs associated with CPE, although we were unable to generate gene knockout mutants.

- Most importantly, the genetic approach is lacking reconstituted controls for the gene deletions, and does not compensate for possible off-target effects through CRISPR. Here, genetic replicates (figure S6) are the only control. I am not sure if this is up to standards, please discuss this in a potential rebuttal.

We understand the reviewer's concern about the lack of reconstituted controls for the gene deletions and the potential for off-target effects through the CRISPR method we used. Concerning the reconstituted controls for the gene deletions, as the reviewer notes, we opted instead to analyze the phenotypes of independent transformants (Figure S6). Figure S6 shows the growth patterns of 2 independent Δ Afu3g13230 gene deletion mutants and 3 independent Afu4g07080 gene deletion mutants in three caspofungin concentrations (0, 0.125 and 4 ug/ml). The growth patterns are highly similar between replicates which confirms the observed phenotypes.

Concerning the off-target effects of CRISPR, the Fortwendel lab demonstrated the CRISPR/Cas9 system we used in this manuscript is not associated with increased mutations linked to the Cas9 nuclease (<https://doi.org/10.1186/s40694-018-0057-2>). The Fortwendel lab has used this approach to generate hundreds of mutants in *A. fumigatus* and have not observed off-target issues.

- The study also relies on the assumption that CPE is a gain-of-function phenotype, it might just as well be the opposite, so that continuous (over)expression might actually show the expected phenotype, or its loss (depending on the perspective and strain used). SNPs in potential regulatory regions (transcriptional upregulation of Afu3g13230 in CEA17!) are completely ignored. Also the transcriptional measurements are not discussed further, but might help the reader.

We understand the concern raised by the reviewer. However, we did not make any assumptions regarding whether the CPE is a gain-of-function or loss-of-function

phenotype. For instance, the only SNP associated with CPE in Afu4g07080 was present in the upstream region. Afu3g13230 also contained a significantly associated SNP that was in the upstream region.

Guided by the reviewer's comments, we tested whether overexpression of Afu3g13230 or *dscP* resulted in the CPE phenotype, or an exaggerated CPE phenotype. We generated 2 independent overexpression mutants for Afu3g13230 and three independent overexpression mutants for *dscP*, by replacing the exogenous promoters with the *hspA* promoter (as described here: <https://doi.org/10.1128/mBio.00437-19>). The overexpression mutant phenotypes were identical to the wild type (i.e. maintenance of the CPE phenotype), suggesting the expression of these genes are required for CPE. We have added this information to the manuscript and have included a new supplementary figure that shows the phenotypes of the parental strain, a gene deletion mutants and overexpression mutants (Figure S5).

The gene expression values we present in Figure S9 were gleaned from a previous study by the Goldman lab in which the CEA17 strain, which displays the CPE phenotype, was grown in the presence and absence of caspofungin (<https://doi.org/10.1128/mBio.00816-20>). The observation that expression was higher in Afu4g07080 and Afu3g13230 during exposure to caspofungin, and that the CPE phenotype is lost in the gene deletions, suggests expression of these genes are necessary for CPE. We have included mention to these results to the discussion.

- Since the authors are using CRISPR, it would have been a more informative approach to test these individual SNPs directly, instead of modifying the entire region by deleting larger chunks (aka genes).

We understand the reviewers concern. It is important to note that genome wide association is an approach to find regions linked to particular phenotypes and, in most organisms and systems, it is rare to identify the specific allele associated with phenotypes. The SNPs we identified may not be causative, but may be linked to causative variants that would be difficult to detect with short-read DNA sequencing. We are unable to characterize certain structural variants that may be causative (i.e. large indels, tandem repeats, translocations, inversions etc.). For these reasons we opted to generate gene deletion and overexpression mutants, rather than explicitly test individual SNPs. Although this is beyond the current scope of this study, we hope to generate long-read sequencing genomes for a closely related subset of strains with and without the CPE phenotype in an effort to further pinpoint the genetic variants that contribute to CPE.

Additionally, guided by the reviewer's comment, we modeled a portion of the Afu3g13230 protein that contained the two missense variants associated with the CPE phenotype (Figure S9). We show that one of these missense variants is located within a highly structured region of the protein which has potential

functional implications. We feel this analysis has strengthened the discussion of this candidate gene.

- In summary, as far as the wet-lab experiments in this study go, they are sound and the implication of the two genes in CPE is derived from solid data. However, there might be more potential in the data and it the study could benefit from a more in-depth analysis of individual SNPs in both genetic directions.

We thank the reviewer for their constructive feedback. We have significantly revised our manuscript highlighted by:

- (1) A supplementary figure that provides a detailed schematic of the chromosome 3 locus (Figure S5).**
 - (2) Generation and phenotypic analysis of Afu4g07080 and Afu3g13230 overexpression mutants (Figure S8).**
 - (3) Protein modeling of Afu3g13230 to provide further insight into the putative implications of missense variants associated with CPE (Figure S9).**
- I have no previous expertise with GWA studies, and can therefore not judge the bioinformatics approach used to determine SNPs and their ranking.

Text and formatting issues:

- Results obtained for gene deletions outside chr3 are omitted in the discussion, please include at least one sentence

We only discuss the results for Afu3g13230 and *dscP* as the gene deletion mutants showed phenotypic differences compared to the wild-type. We added a sentence at the end of the second paragraph in the discussion noting this observation.

- Lines 124-134 could be removed: there is no doubt GWA is a useful method in fungi

We have opted to keep this paragraph in our manuscript. Our goal with this text was to highlight the potential for GWA in fungi, as it is currently an underutilized approach. We have shortened this section.

- Lines 138ff: create a better list of concentrations, e.g. "without and with 0.125, 0.25, 0.5, 1, and 8 mg/l, respectively.

We made this modification.

- Or similar Paragraph with accession numbers: please integrate numbers into table S1. Use the space to briefly describe the sequencing technology used in the references 11 and 55 instead.

We opted to keep the accession number paragraph in the main text and we added the accession numbers to Table S1. As noted in the first sentence of the

“*Aspergillus fumigatus* whole-genome Illumina data” section of the methods, all isolates have accompanying whole-genome paired-end Illumina data.

- Line 214 and throughout text: replace "knockout" by "gene deletion" or "gene inactivation" as appropriate.

We have replace knockout with gene deletion.

- Line 262: remove "(see methods)"

We removed this text.

- Figure 3 gave me a lot of trouble in the pdf because the dots are all individual objects. Please make sure that another figure format is chosen (e.g. tiff)

We apologize for this. We have converted the Manhattan plots in figure 3 to tiff format.

- Figure4. move label for dAfu3g13270 to right. Add human-readable gene names to make interpretation easier, where available.

We have aligned the Afu3g13270 label and added gene names where applicable.

- Figure 5B/C and 6A: in the pdf sometimes there is a rho instead of a delta preceding the gene names (where I think a delta should be). Please change, or elaborate (perhaps I am just not familiar with that nomenclature)

We apologize for this formatting issue. We have changed the rho symbols to delta symbols.

- Table S1, in addition to the adding accession numbers, pleas rename the last column to something that makes clear this is not the reference for the phenotypic data, but only for the strain source. Remove concentrations from table legend, as they are also in the table.

We have incorporated these suggestions.

Reviewer #2 (Comments for the Author):

- This work focused on the "Caspofungin Paradoxical Effect" and searched for causative genes by genome-wide association (GWA) analysis. And, two genes were shown to contribute to CPE. The methods and conclusions are scientifically sound. I would like to make a few comments, which I hope you will find useful.

Comments

- Introduction section Line 87 "while azoles disrupt ergosterol biosynthesis resulting in toxic sterols (31)." This statement is misleading. This is one hypothesis, so please rephrase it.

We modified this phrase to “while azoles interfere with the biosynthesis of ergosterol”.

- Resulte, CRISPR/Cas9 gene deletion of candidate genes, section Line 321-322 "~ a putative diacylglycerol kinase (detected in all GWA), DspC, a predicted tyrosine phosphatase (87) (detected in 67-sample GWA), Afu7g01440,~" ; Please correct the following statement. "~ a putative diacylglycerol kinase (detected in all GWA), Afu4g07080, which encodes DspC, a predicted tyrosine phosphatase (87) (detected in 67-sample GWA),~"

We made this correction.

- And, it would be helpful to add a table with gene ID, gene name, and protein to the supplemental.

We only mention two genes with different gene names/gene IDs (Afu4g07080, *dscP* and Afu3g13270, *dgkA*) in the manuscript and therefore opted not to add additional information to the supplement.

- Two names are used for one mutant like $\Delta dscP$ and $\Delta 4g7080$. Please unify either of them.

We have changed instances of $\Delta Afu4g07080$ to $\Delta dscP$.

- Discussion section Line 421 and 424; There are two designations, WT CPE+ isolate and CPE+ WT isolate. Do these refer to the same?

We have corrected this typo and use WT CPE+ in both instances.

- The data presented in Table S1 shows that the strains that have a Recovery Rate of 0 are growing well. This is different from the phenotype observed for Afu3g13230 or *dspC* deficient strains. It would be better to discuss this.

We appreciate the reviewer’s comment regarding the reduced growth rate in gene deletion mutants (Afu3g13230 and Afu4g07080) in the absence of casopfungin. The gene deletion mutants likely do not represent the mechanisms by which alleles in these genes are affecting the CPE phenotype. For instance, Afu3g13230 contains two missense variants, and differences in protein structure, rather than null expression, may influence the CPE phenotype. Additionally, as mentioned above, GWA is helpful in identifying variants associated with CPE but we would need contiguous assemblies of CPE+ and CPE- strains and additional molecular genetics to comprehensively characterize the causative variants. The gene knockouts do implicate the involvement of Afu3g13230 and *dscP* in the CPE phenotype, but we speculate the gene deletion phenotypes differ from the CPE phenotypes because the mechanisms differ (i.e. protein structure or change in

gene expression rather than null expression). We have added some discussion of this to the discussion section.

- For the two genes you focused on in this work, have you examined the gene expression levels et.al. in the CPE+ isolate shown in Figure 1? There is little discussion of the contribution of the two genes to CPE in these isolates.

Unfortunately, we do not have direct gene expression data for Afu3g13230 or *dscP* for the strains in Figure 1. However, as displayed in Figure S9, we report expression values from a previous study in the Goldman lab (<https://doi.org/10.1128/mBio.00816-20>) for CEA17 (which displays the CPE phenotype) in the absence of caspofungin and in the presence of 2 μ M caspofungin, for the 7 genes we constructed gene deletion mutants for. In the results section we note that Afu3g13230 and *dscP* both show higher levels of expression in the presence of caspofungin. However, since we have not surveyed the transcriptional profiles of numerous CPE+ and CPE- strains, we did not give the expression data much attention in the discussion.

August 17, 2022

Dr. John G Gibbons
University of Massachusetts Amherst
Food Science
240 Chenoweth Lab
102 Holdsworth Way
Amherst, MA 01003

Re: Spectrum00519-22R1 (Genomic and molecular identification of genes contributing to the caspofungin paradoxical effect in *Aspergillus fumigatus*)

Dear Dr. John G Gibbons:

Your manuscript has been accepted, and I am forwarding it to the ASM Journals Department for publication. You will be notified when your proofs are ready to be viewed.

Please address the final comments regarding the addition of standardized typing data made by reviewer #1 in your final proof.

Sincerely,

Rhys Farrer
Editor, Microbiology Spectrum

Journals Department
Supplemental Material FOR Publication: Accept